# Feasibility of Old Bark and Wood Waste Recycling

**DOI:** 10.3390/plants11121549

**Published:** 2022-06-10

**Authors:** Yuliya Kulikova, Stanislav Sukhikh, Olga Babich, Margina Yuliya, Marina Krasnovskikh, Svetlana Noskova

**Affiliations:** 1Institute of Living Systems, Immanuel Kant BFU, 236016 Kaliningrad, Russia; stas-asp@mail.ru (S.S.); olich.43@mail.ru (O.B.); svykrum@mail.ru (S.N.); 2Environmental Protection Department, Perm National Research Polytechnic University, 614000 Perm, Russia; makarova_u85@mail.ru; 3Department of Inorganic Chemistry, Chemical Technology and Technosphere Safety, Perm State National Research University, St. Bukireva, 15, 614990 Perm, Russia; krasnovskih@yandex.ru

**Keywords:** biomass, bark waste, composting, hydrothermal methods, biofuel, sustainable resources, wood forest product, lignin

## Abstract

The pulp and paper industry leads to the formation of significant amounts of bark and wood waste (BWW), which is mostly dumped, causing negative climate and environmental impacts. This article presents an overview of methods for recycling BWW, as well as the results of assessing the resource potential of old bark waste based on physicochemical and thermal analysis. It was found that using BWW as a plant-growing substrate is challenging because it was observed that bark waste is phytotoxic. The C:N waste ratio is far from optimum; moreover, it has a low biodegradation rate (less than 0.15% per year). The calorific value content of BWW ranged from 7.7 to 18.9 MJ/kg on d.m., the ash content was from 4% to 22%, and the initial moisture content was from 60.8% to 74.9%, which allowed us to draw conclusions about the feasibility of using hydrothermal methods for their processing to obtain biofuel and for the unreasonableness of using traditional thermal methods (combustion, pyrolysis, gasification).

## 1. Introduction

The Russian Federation is ranked second in the world for wood reserves and sixth for wood processing [1]. The problem of effective and integrated use of wood waste is becoming increasingly urgent as the volume of its processing increases. The pulp and paper industry waste, primarily BWW and dehydrated cellulose sludge, contributes significantly to the composition of wood waste [2].

The volume of BWW generation in Russia is approximately 2 million tons [3], with only 2% being processed [4]. In comparison, the United States pulp and paper industry recycles approximately 20 million tons or 25% of BWW per year [5,6].

BWW is not considered to be a waste in countries with well-developed forestry, such as Finland, Sweden, Germany, Canada, and the United States, but rather as a raw resource from which valuable products can be obtained. This approach not only reduces the cost of production, but also brings additional profit to the pulp and paper industry [7]. BWW is already widely used in bioenergy in developed countries [8,9,10]. China is also on the bioenergy path, with plans to produce 50 million tons of fuel pellets per year by 2025 [11,12].

Russia is now in the process of stimulating BWW processing and recycling. Therefore, one of the goals of the Russian Federation program “Development of industry and increasing its competitiveness” is to develop the production of bioenergy and biofuels and increase the processing of low-grade wood and bark waste [13]. However, only a small amount of wood debarking waste is currently used for incineration and agricultural purposes in Russia [4], with the majority of waste being disposed of in landfills.

Many bark dumps today contain tens of millions of cubic meters of BWW, which is rarely used for commercial or energy production but causes extensive environmental damage. An ecological threat is posed by the acidity of the soil and contamination of water bodies with extracts and products of bark decay; moreover, BWW forms large amounts of greenhouse gases, and the dumps are highly inflammable during arid periods [3,13].

Natural biodegradation in real conditions occurs much more slowly than previously thought, particularly in flooded layers [13,14,15,16]. It was shown that even after more than 80 years of storage, complete humification of wood in the body of bark dump does not occur (wood chips and bark particles completely retained their structure in samples of 75–82 years storage). The preceding highlights the importance of creating solutions for the safe disposal and use of BWW, particularly those that have been stored for a long period [17].

Summarizing global experience, various options for using BWW from the pulp and paper industry are currently available, which can be classified into three categories: chemical, biological, and thermal.

The main chemical processing methods for BWW are extraction and hydrolysis. Organic solvents (hexane, isopropanol) and water are used as extractants [18]. The hydrolysis of BWW in the presence of catalysts (acid salts, mineral acids) produces a variety of food, feed, and industrial products (alcohol, yeast, carbon dioxide) [19]. The scientific literature lacks information on the experience of the use and proven effectiveness of chemical methods in relation to long-stored bark.

The biological method of BWW processing is the most widely used, especially for long-term storage waste. It is based on anaerobic digestion and composting processes [3,20,21,22]. Composting can be carried out both in the field and in various types of bioreactors (bio-drums, biotunnels, etc.) in order to produce fertilizers and ameliorants [23]. Preparing BWW for biodegradation involves preliminary grinding and the addition of various additives, depending on the intended type of target product. BWW is pre-treated with urea, calcium carbonate, phosphate, and zeolite to concentrate nitrogen and phosphorus, and the resulting material is placed in compost heaps. Lime or ash can be added to the compost to adjust the pH of the final product [3,24,25,26,27].

Products can be used for intermediate waste isolation on landfills, reclamation of disturbed lands, landscaping, or as fertilizer to improve the topsoil structure and to stimulate the growth of plants [2,3,28,29,30].

Biotechnological approaches to the pulp and paper industry’s waste management, based on biocatalytic and biotransformation processes, are currently in demand [4]. Micromycete fungi that produce extracellular enzymes, various plant metabolites, biopolymers (cellulose, hemicellulose, lignin, humus) [31,32], and xenobiotics [33] are the cultures suitable for processing pulp and paper industry waste components. Various bacterial cultures, including biodestructors of plant components, and such worm cultures such as *Eisenia fetida*, *Eisenia andrei*, *Eudrilus eugeniae*, *Perionyx excavates*, and *Perionyx sansibaricus* [24,25] can also be used. The product of vermicomposting, or biohumus, obtained from organic waste, undergoes physicochemical, biological, and microbiological transformations in the intestines of worms to obtain a granular structure [3,25,34]. However, these technologies are still under development and have not been widely used.

Thermal methods are represented by combustion, pyrolysis and gasification, and hydrothermal methods [35,36,37,38]. Thermal or electrical energy is the primary product of BWW combustion; ash (10–20% by mass) is formed as a waste [39]. The pyrolysis process generates thermal energy, but in the case of BWW processing, it is typically fully utilized to support the process. Charcoal is a byproduct of the pyrolysis process that can be used commercially as a sorption material, biofertilizer, and for greenhouse gas sequestration [39,40,41,42,43]. Dry gasification of BWW is used to generate heat and electricity, as well as for the synthesis of gas [40]. The problem with all traditional thermal processing methods is that they are difficult to apply to the processing of BWW previously accumulated in the bark dump, because the waste is characterized by high humidity (up to 60%) [14,44].

Hydrothermal methods, which ensure the processing of wet biomass without pre-drying, are a promising branch of thermal methods for BWW processing. Hydrothermal carbonization (HTC) is the process of converting cellulosic low-calorie biomass (with humidity up to 80%) at temperatures of 250–300 °C and at pressures of 2–20 MPa into hydrochar [32,45,46]. The yield is 62–78% of dry biomass, which can be further used to generate heat as a carbon source [47]. Temperature (optimally not less than 250 °C), water:substrate ratio (optimally no more than 1:4) [48], and pH (optimally no higher than 3) all affect the efficiency of the process [14,48]. The typical time of the HTC process is from 1 to 72 h [49].

Hydrothermal liquefaction (HTL) is a wet-biomass processing method that produces liquid synthetic oil [18,35,44]. The process is designed to treat pulp with a maximum dry matter concentration of 20% [50]. The efficiency of the process is largely affected by the pH value; thus, with a decrease in pH, the yield of char increases significantly, and the yield of liquid fuel decreases [44,51]. The yield of synthetic oil for various types of wood is 10–30% [52,53,54]. For example, the organic components of HTL-oil derived from BWW have an average molecular weight of 310–470 g/mol [55] and are mainly represented by carboxylic acids, furfurals, ketones, aromatic, saturated and unsaturated hydrocarbons. HTL-oil is viscous and, unlike pyrolysis oil, has a low oxygen content and a higher calorific value [36]. The gas and water phases, as well as HTL-char [44], are the byproducts of the HTL process.

Hydrothermal gasification of BWW is carried out at a temperature of about 350 °C and a pressure of 20 MPa, resulting in the formation of a methane-rich gas [40]. The ideal conditions for hydrothermal gasification include a biomass dry matter concentration of less than 10% [40], a high heating rate, the addition of alkali salts (K_2_CO_3_, KHCO_3_, etc.) to prevent coke formation [56], and temperatures of 400–550 °C [57,58]. There is experience in implementing processes at lower temperatures (270–450 °C) in the presence of homogeneous and heterogeneous catalysts [51,59].

The general advantage of hydrothermal methods is their applicability to wet biomass, because water does not interfere with the process, but participates by acting as a donor of hydrogen ions and, in some cases, as a polar solvent [44]. Although HTC is the simplest and most easily implemented method, the resulting hydrochar is not widely sold. More valuable products are syngas/hydrogen and liquid fuels. The processes of HTG of low-calorie wastes are beginning to develop, and this process is rather complicated because wastes are characterized by significant heterogeneity [60]. In this regard, HTL with liquid fuel production appears promising. The efficiency of hydrothermal processes significantly depends on the calorific value, ash content, and elemental composition of the biomass. It is obvious that physical and chemical analyses are important for assessing the possibility of processing long-term stored BWW. This article aims to characterize the physicochemical and elemental composition, as well as the thermal properties of BWWs of various storage times, in order to establish the most promising direction for their use.

The novelty and practical value of this work lies in the fact that, for the first time, the properties of BWW of various storage periods are compared, the most common methods of processing and recycling BWW are summarized, as well as the feasibility of using hydrothermal methods and composting for BWW processing.

## 2. Materials and Methods

Sampling of BWW was conducted from the bark dump body (city Krasnokamsk, Russia), which currently contains 1.5 million m^3^ or 1.2 million tons of BWW and which occupies an area of 22.3 ha (Figure 1).

The accumulation of BWW took place between 1936 and 2005 (until the closing of the pulp and paper company). The industrial waste disposal facility is located at a distance of 120 m from the residential area of Krasnokamsk city, mostly on the territory of the coastal protection belt of the Votkinsk reservoir. The height of the dump varies from 2 to 21 m.

Sampling was carried out by drilling at three points. Drilling was performed using a mechanical core method using the URB-2A rig (Mashinostroitelnyy zavod im. V.V. Vorovskogo, Yekaterinburg, Russia). An auger-type drill with a diameter of 127 mm was used (Figure 2). When choosing sampling points, the goal was to select wastes characterized by different ages of being in the bark dump body.

Hydrogen index and chemical oxygen demand (COD) were measured in water extract. An aqueous extract was prepared as follows: 5 g of bidistilled water was taken for 1 g of waste and shaken for 5 min. The resulting suspension was filtered through a white tape filter.

The hydrogen index was determined by the potentiometric method using the Expert pH tester (Ekonis-Ekspert, Moscow, Russia).

The COD was estimated based on ISO 6060:1989 by the method of oxidation of organic compounds with potassium dichromate in an acid medium at boiling, followed by titration of the residual amount with Mohr’s salt [61].

Humidity was determined gravimetrically by drying at 104 °C to constant weight. Loss on ignition (LOI) was also determined gravimetrically by calcination at 550 °C, similar to the method described in ASTM-D7348 [62].

Respiration activity was assessed in accordance with OENORM S 2027-4:2012 “Evaluation of waste from mechanical–biological treatment. Part 4: Stability parameters—Respiration activity (AT4)” [63].

Elemental analysis of algae biomass was performed using a CHNS elemental analyzer Elementar Analysensysteme (Germany) model Vario EL Cube. Weighing was carried out on an analytical balance with an accuracy of 0.01 mg. The content of elements was determined based on the area of the chromatographic peaks of N_2_, CO_2_, H_2_O, and SO_2_ using a calibration straight line constructed using standard compounds. Each sample was examined in three separate repetitions, with mean values reported. The analytical data were processed, and the content of components in the sample was calculated using the software provided by the equipment manufacturer.

To evaluate the calorific value and thermal properties of BWW samples, a simultaneous thermal analysis was carried out in oxidizing (air) and inert (argon) media. The studies were carried out on a NETZSCH STA 449C Jupiter synchronous thermal analyzer (NETZSCH-Gerätebau GmbH, Germany). The analysis parameters are shown in Table 1.

## 3. Results and Discussion

### 3.1. Evaluation of Agricultural Potential of BWW

The results of the physicochemical analysis of BWW are presented in Table 2 and Figure 3. Waste age assessment was performed on the basis of the following data: technological plan and register of bark dump filling for the period: 1950–2005.

The waste material sampled from the bark dump had a high moisture content ranging from 60.79% to 74.95%, with an average value of 68.65%. The analysis of the results concludes that the quality of BWW in dumps changes insignificantly over a long period. The pH value does not change significantly and shifts toward neutral values from 6.5 (weighted average for waste with storage period less than 40 years) to 7.5 (weighted average for waste with storage period more than 40 years).

This fact supports the hypothesis that there are no humification processes in the bark dump, because it is accompanied by a pH shift to the acid side, which we did not observe during the physicochemical properties of the waste analyzing. At the same time, we did not observe a significant change in the amount of organic compounds, since the LOI in long-stored BWW differs from the average value for new waste by no more than 5% (the values for new waste and waste with 80 years of storage are 92% and 87%, respectively). The carbon content in BWW stored for over 80 years was 48.13%, which practically does not differ from its content in waste stored for 10 years, which was 47.84%.

With the sufficiently high carbon content, the low level of respiratory activity was observed; thus, the average value of the AT_4_ parameter was 3.6 mg O_2_/kg. This ratio indicates that the contained organic molecules have a low potential for destruction, which could be due to their low bioavailability or to the presence of toxicants.

Thus, we can conclude that organic matter mineralization in the bark dump proceeds slowly. Retardation of mineralization is related to anoxic conditions and to a high concentration of organic compounds with bactericidal properties. Anaerobic communities of microorganisms are especially sensitive to the effects of the toxicants. At the same time, it was not possible to identify the patterns in sulfur and nitrogen content change in waste samples from different storage periods, which is related to the different types of raw materials used (at different times, coniferous or birch wood was used in the technological process), as well as possible contamination with waste pulping, which is rich in sulfur and nitrogen.

A slight decline in the amount of hydrogen and oxygen can be observed (Figure 3c,e). The median value of hydrogen content decreased by 9.8%, with oxygen decreasing by 17.5% in the old BWW. This is associated with the partial destruction of readily available organic compounds in anaerobic processes, accompanied by the formation of water, carbon dioxide, methane, and hydrogen.

The most comprehensive studies of changes in the elemental composition and pH of bark were carried out by a group of scientists on the example of bark decomposition in natural conditions of a boreal forest. The authors also found that the carbon concentration in the spruce bark remained virtually unchanged (spruce BWW dominates in the dump under study) [67]. The authors also noted that the pH of bark and wood waste remained practically unchanged for more than 66 years [67].

The ratio of carbon and nitrogen (C:N) is used as integral characteristic of plant-growing substrates. Under normal conditions, this ratio ranges between 8 and 12, with a deviation indicating the soil’s ecological unfavorability. A deviation in the range of 7.5–20 is considered acceptable [68].

Large deviations are detrimental to soil microorganisms. Almost all of them are sensitive to the carbon–nitrogen balance (C:N). Its deviation in any direction from the specified norm indicates the suppression of soil microbiological processes. If the substrate has a low C:N ratio, then ammonia accumulates in the soil substrate, since microorganisms do not have enough carbon-containing compounds to assimilate nitrogen. If, on the contrary, this ratio becomes high, microbial communities are severely deficient in nitrogen [69].

According to compost standards [70,71], the most favorable C:N ratio in composting substrate is 30:1, and this ratio during the composting process changes to 20–18:1. In the BWW, the C:N ratio varies over a wide range from 34 to 306 (with a zonal norm of 11). The obtained results are consistent with the data on the ratio of carbon and nitrogen in the bark of coniferous trees, reflected in the literature [67,72]. Thus, we see that the lack of biologically available nitrogen also inhibits the BWW composting process. The lack of nitrogen during BWW composting could be replenished by mixing with nitrogen-rich substrates (e.g., manure) or by adding mineral forms of nitrogen (e.g., carbonic acid amide).

### 3.2. Evaluation of BWW Thermal Properties

To evaluate the thermal properties, BWW samples were analyzed using simultaneous thermal analysis in oxygen and inert gas environments (Table 3). An example of a thermogram in an oxygen environment for nine-year-old waste is shown in Figure 4.

Absolutely all samples demonstrated two-stage degradation under oxygen conditions with a maximum average degradation rate of 20–27%/min at a temperature of 330–349 °C. The first peak at temperatures up to 120 °C is associated with the loss of moisture in the samples.

The second peak should be associated with the destruction of the bulk of the organic compounds of hemicellulose and cellulose. The second peak has a shelf in the temperature range of 378–538 °C (average 422.8 °C). This shelf should be associated with the end of the processes of lignin destruction and the combustion of previously formed char.

Barta-Rajnai et al. also noted a two-stage decomposition of the bark in the temperature range of 250–450 °C, with a maximum ratio decomposition at a temperature of 380 °C [66]. However, the HHV established by Barta-Rajnai et al. for the Norway spruce bark was 20.14 MJ/kg, which is 35.6% more than what was obtained in our studies (average value for BWW). This is most likely due to the fact that fresh bark was analyzed in the Barta-Rajnai et al. studies and not old BWW, as in our studies.

Figure 5 depicts a comparison of the dynamics of degradation in an oxygen environment of BWW samples of different storage periods. Figure 5 shows an array of temperature values at which the maximum rate of samples mass loss was observed; this is the extremum of the curve “mass loss/time”.

The lower the temperature where this extermum is observed, the less thermally stable the sample and the lower its degree of mineralization. Obviously, as the storage period increases, the temperature at which we see the most active decomposition of the sample increases, which is related to partial hemicellulose degradation and biomass mineralization.

According to the literature data, the bark of coniferous trees contains 16–23% cellulose, 13–31% hemicellulose, 8–10% polyuronides, 27–33% lignin, and 14–30% extractives [73]. Simultaneously, during storage, wood loses some hemicellulose, which is partially depolymerized when soluble, biologically readily available components are eliminated (mainly mono- trisaccharides). As previously stated, the successful destruction of BWW is significantly hindered by its constituent tannin and guaiacyl lignin. The latter is less prone to degradation compared to conventional lignin, since it contains fewer aryl–aryl bonds and, as a result, a lower redox potential [74,75].

At the same time, we can observe a drop in the total calorific value of BWW (on average by 12 percent) when we examine the thermal effects that accompany the processes of thermal destruction of samples (Figure 6).

The calorie content of the waste directly depends on the content and form of organic compounds of the biomass. Due to the destruction of BWW organic matter, the ratio of organic/inorganic substances in the samples changes, and as a result, we can observe a decrease in HHV (Figure 6) and increase in ash content of the biomass (Table 4). These data indirectly confirm partial mineralization of the organic matter. However, waste mineralization is slow, with only 0.15% of organic matter being mineralized per year. This fact is also confirmed by a slight increase in the ash content of waste over 75 years (by no more than 5.4%).

If we look at the curve of samples mass loss rate in an argon atmosphere, we can clearly distinguish only one stage of thermal destruction, described in Table 2. The absence of the second stage of destruction in argon and the presence of this stage in air is due to the fact that the second stage characterizes the oxidation of pyrocarbon formed in the previous stages, and this process is impossible without oxygen. However, on the DSC curve, we could recognize a bit more peaks. The first two peaks on the DSC curve (280–380 °C and 380–475 °C) most likely correspond to the autoxidation of hemicellulose and cellulose. Furthermore, at temperatures as high as 660 °C, the additional destruction of inorganic matter can be observed. The maximum degradation rate was observed at a temperature of 357–380 °C and was 9–12% per minute. The residual amount of pyrocarbon and ash residue varied significantly within 24.8–48.2%. It was not possible to establish a pattern indicating the relationship between the storage period of waste and the proportion of pyrolysis residue. An example of a thermogram in an argon environment for a sample of 82 years of storage is shown in Figure 7.

To evaluate the feasibility of using BWW as a fuel source, it is worth comparing it with the well-known biomass sources that are actively used as solid fuels or for the production of liquid hydrocarbons. Coniferous wood, biomass of macro- and microalgae, and straw are the closest analogues. A comparison with the above-mentioned types of biomass is presented in Table 4. For BWW in Table 4, we used the average values of the main elements, HHV and ash content.

Due to the low calorific value and high moisture content, BWW biomass has a low potential attractiveness as a solid fuel for direct incineration without pre-conversion. The low calorific value of the waste, taking into account the natural humidity at a level of 68.6%, was 4.49 MJ/kg. The low sulfur and nitrogen concentrations of BWW permit conclusions to be drawn about its prospective applicability for producing liquid and solid fuels in hydrothermal conversion processes.

The preliminary treatment of bark biomass by hydrothermal methods will provide an increase in the specific calorific value of biomass due to the autoxidation of a part of the oxygen-containing compounds and the transition of their conversion products to the liquid phase [83].

## 4. Conclusions

Processing of primary raw materials on the pulp and paper enterprises gives large volumes of wood waste, consisting mainly of bark (60–70%). Up to 98% of such waste in Russia is landfilled in the environment without any pre-treatment. BWW disposal in bark dumps creates risks of an uncontrolled release of pollutants into the environment [2].

Analysis of the physicochemical properties of BWW on the example of waste from a pulp and paper plant located in the Ural region of Russia (Krasnokamsk city) allowed us to establish that even long-stored waste (80 years or more) does not undergo significant changes during storage in a bark dump. This fact is confirmed by only slight decreases in the LOI (by no more than 5% over more than 80 years of storage) and the specific calorific value of waste (on average by 12%). Mineralization of BWW proceeds slowly (the calculated rate of destruction of organic matter is no more than 0.15% per year), which is associated with the presence of a number of biodegradation inhibitors in the waste, in particular, tannin and guaiacyl lignin. Therefore, the waste of bark dumps for a long time (more than 200 years) will not be significantly subjected to the processes of natural biological destruction, which suggests the relevance of finding ways to rationally process BWW.

An evaluation of the potential for using BWW to produce plant-growing substrates and fertilizers revealed that the waste has a low nitrogen content (average C:N ratio 119) and a high content of difficult-to-oxidize organic matter, which is supported by the following data: with an average content of organic compounds in waste of 91.9% (mean LOI), mean respiratory activity does not exceed 4 mgO_2_/kg d.m. Thus, BWWs can only be used as a bulking agent when preparing soil mixes with other organic wastes rich in biogenic components (for example, manure). At the same time, preliminary testing to determine their phytotoxicity is essential, because lignin and other bark components might hinder plant development [84].

The thermal properties of BWW were studied, and it was discovered that the waste has a high humidity (61–75%) and a low calorific value (14.33 MJ/kg on d.m. and LHV = 4.5 MJ/kg). This means that the use of traditional thermal methods (pyrolysis, incineration, gasification) will require pre-drying of the waste, resulting in a negative or inefficient energy balance.

At the same time, this waste is characterized by a low ash content (average 8.1%) and high carbon content (average 46.4%), with low contents of nitrogen (average 0.4%) and sulfur (average 0.3%). This allows us to conclude that hydrothermal conversion methods, such as hydrothermal carbonization (HTC), hydrothermal liquefaction (HTL), and hydrothermal gasification (HTG), are promising for the processing and utilization of large volumes of BWW. The high content of carbohydrates in the form of cellulose and hemicellulose and the low content of proteins and fats make it possible to draw conclusions about the advisability of mixing this type of biomass during hydrothermal liquefaction processes with other sources of biomass that are characterized by a high content of proteins (for example, algae). This will achieve a synergistic effect due to the synthesis of ketosamines and their products of further transformation according to Maillard reactions between carbohydrates and proteins [50].

Based on this research, it was found that the most promising directions for the utilization of old BWW should be considered, such as their use as structurants in the production of compost and BWW hydrothermal conversion to obtain hydrochar and liquid fuel. Further studies on the feasibility of old BWW recycling should be devoted to the evaluation of their phytotoxicity and the dependence of this parameter on the storage period, in case of use for compost production. Development of HTL processing of BWW should follow the path of searching for optimal conditions, including the possibility of co-processing with other organic waste and using catalytic systems.

## Figures and Tables

**Figure 1 plants-11-01549-f001:**
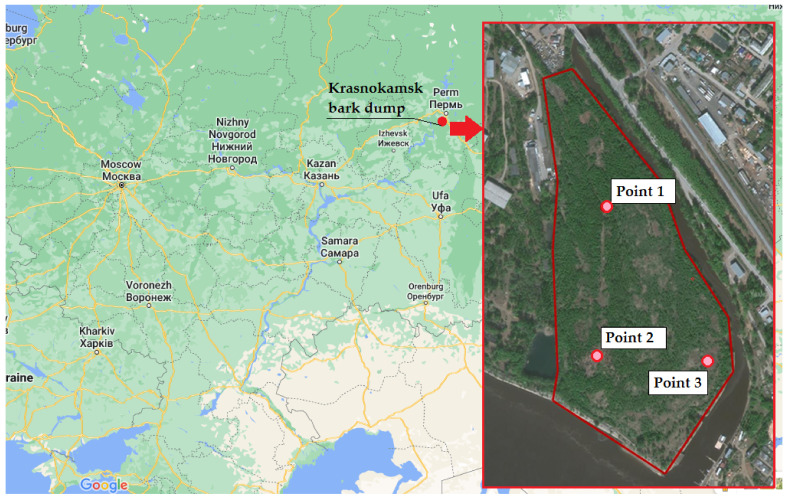
Location of the studied bark dump and sampling site (primary cartographic basis from the website https://www.google.com/maps/, accessed on 18 March 2022).

**Figure 2 plants-11-01549-f002:**
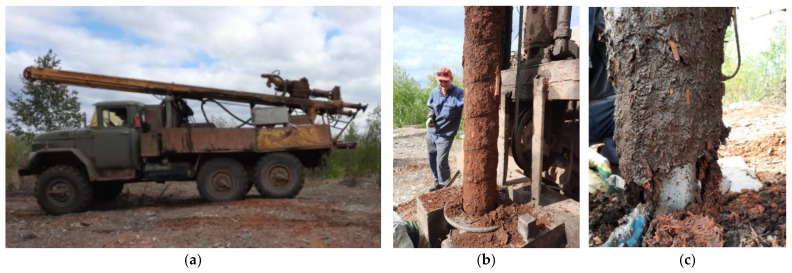
Sampling process: (**a**) URB-2A drilling rig; (**b**) general view of the drilled core; (**c**) BWW condition and structure.

**Figure 3 plants-11-01549-f003:**
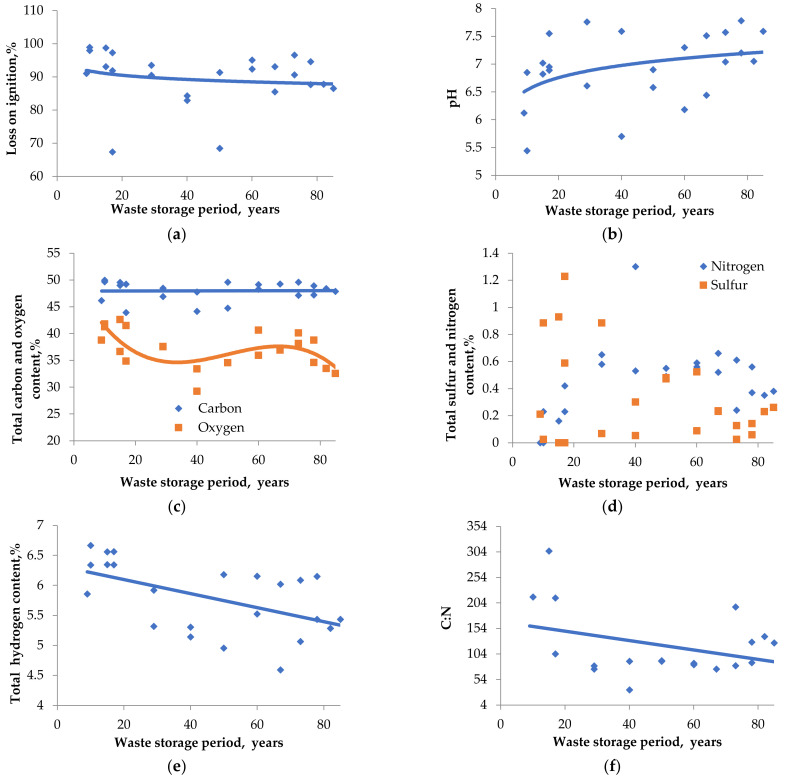
Results of physicochemical analysis of BWW samples: (**a**) loss on ignition; (**b**) pH; (**c**) carbon and oxygen content; (**d**) sulfur and nitrogen content; (**e**) hydrogen content; (**f**) carbon to nitrogen ratio.

**Figure 4 plants-11-01549-f004:**
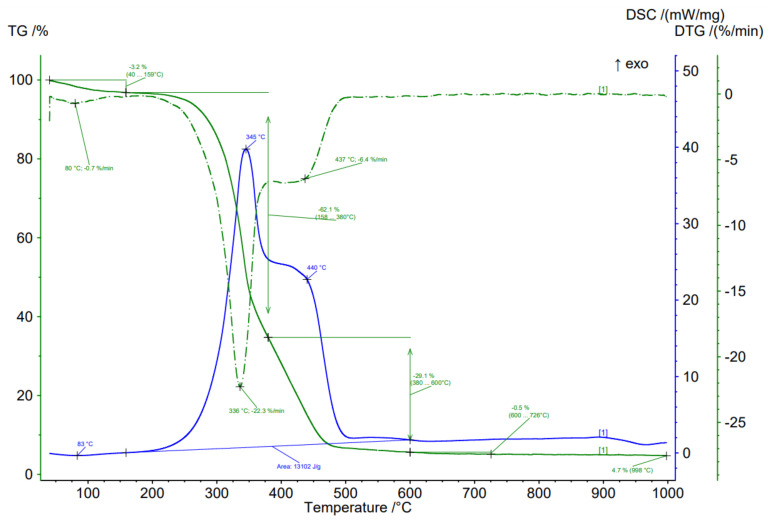
Results of simultaneous thermal analysis in the air atmosphere on the example of BWW with a nine-year storage period.

**Figure 5 plants-11-01549-f005:**
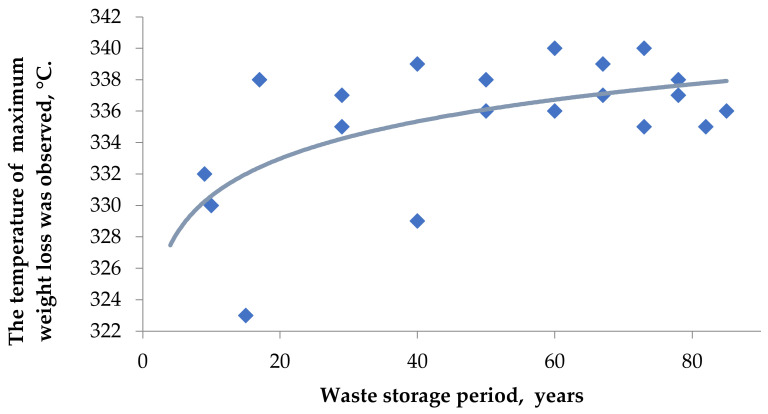
Change in the samples’ thermal stability.

**Figure 6 plants-11-01549-f006:**
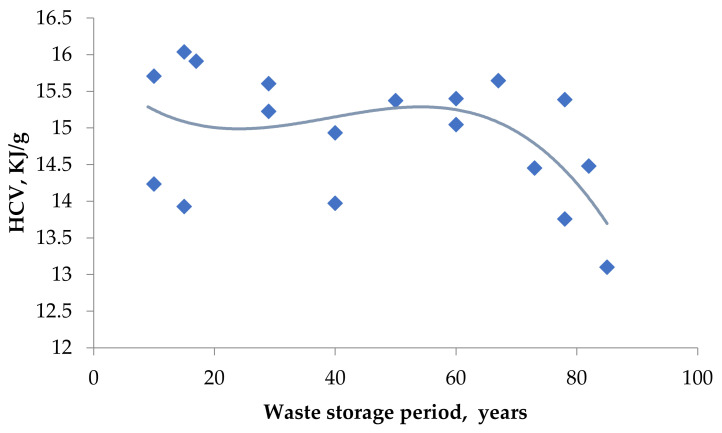
Changing of the BWW high calorific value.

**Figure 7 plants-11-01549-f007:**
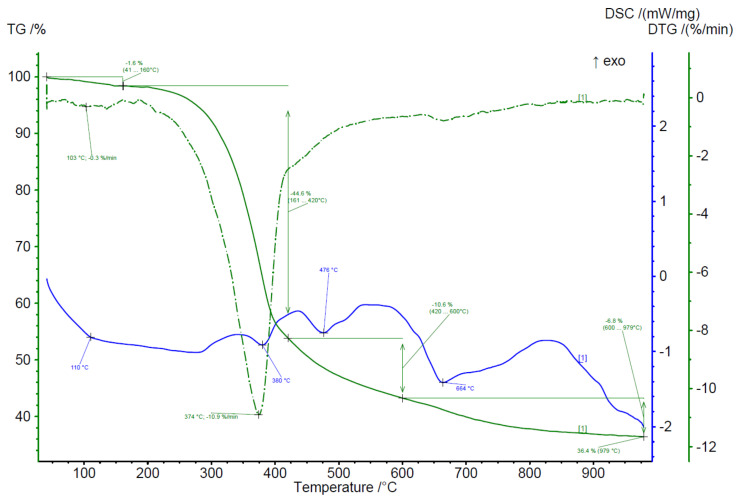
Results of simultaneous thermal analysis in the air atmosphere on the example of BWW with an 82-year storage period.

**Table 1 plants-11-01549-t001:** Simultaneous thermal analysis conditions.

Parameter	Value
Initial temperature:	30/40 °C
Dynamic segment:	1000 °C
Heating rate	20 degrees/min
Furnace gas flow rate	40 mL/min air/argon
Pan	PtRh20 85 µL, with lead

**Table 2 plants-11-01549-t002:** Results of the physicochemical analysis of BWW samples.

Depth, m	Age, Years	Humidity,%	AT_4_, mgO_2_/kg	pH	LOI,%	C,%	H,%	N,%	S,%	O,%	C:N
Well 1											
1	10	62.02	7.89	5.44	97.91	49.96	6.669	0	0.024	41.26	-
3	15	66.26	6.6	6.82	98.71	49.54	6.562	0	0	42.61	-
5	17	62.99	4.3	6.89	97.25	49.21	6.565	0	0	41.48	-
Well 2											
1	9	60.79	3.59	6.12	69.23	46.15	5.858	0	0.211	38.78	-
3	10	65.2	4.1	6.85	98.9	49.67	6.342	0.23	0.885	41.77	216
5	15	67.76	7.83	7.02	93.05	48.97	6.349	0.16	0.929	36.64	306
7	17	69.6		6.95	91.87	49.22	6.345	0.23	1.228	34.85	214
9	29	71.25	4.8	6.61	93.49	48.48	5.922	0.65	0.885	37.55	75
11	40	69.85	1.34	4.70	84.29	44.14	5.143	1.3	0.3	33.41	34
13	50	74.95	4.17	6.58	91.32	49.57	6.183	0.55	0.472	34.55	90
15	60	74.77	3.88	6.18	92.32	49.13	6.154	0.59	0.523	35.92	83
17	67	68.03	7.13	6.44	93.07	49.26	6.02	0.66	0.231	36.90	75
19	73	71.5	1.85	7.04	96.54	49.59	6.089	0.61	0.126	40.13	81
21	78	73	-	7.78	94.55	48.91	6.151	0.56	0.141	38.79	87
Well 3											
1	17	62.72	1.28	7.55	67.33	43.91	2.947	0.42	0.588	22.41	105
3	29	70.91	1.61	7.76	90.44	46.89	5.318	0.58	0.068	37.58	81
5	40	69.49	2.73	7.59	82.87	47.76	5.305	0.53	0.053	29.22	90
7	50	66.08	1.6	6.9	66.93	44.73	4.955	0.49	0.481	17.74	91
9	60	69.68	4.89	7.3	95.07	48.28	5.525	0.56	0.088	40.62	86
11	67	70.38	2.29	7.51	85.47	40.48	4.592	0.52	0.236	80.12	78
13	73	70.15	1.72	7.57	90.59	47.11	5.066	0.24	0.024	38.15	196
15	78	66.53	1.2	7.2	87.65	47.2	5.437	0.37	0.058	34.59	128
17	82	70.58	1.17	7.05	87.74	48.4	5.287	0.35	0.23	33.47	138
19	85	73.07	-	7.59	86.47	47.86	5.435	0.38	0.26	32.54	126
Podzolic soils ^1^ [64,65,66]				5.7	16.0	12.1	n/d	1.37	n/d	n/d	11
Dark humus soils ^2^ [64,65,66]				7.3	49.2	50.8		1.27	n/d	n/d	23

^1^ Average for zonal podzolic soils. ^2^ Average for zonal dark humus soils.

**Table 3 plants-11-01549-t003:** Results of the thermal analysis of BWW samples in the air and in argon.

	Waste Storage Period, Years	Atm.	Number of Main Stages	t_1_	t_2_	tmax	∆m, %	Ash, %	HHV, KJ/g
Well 1	10	O_2_	1	148	372	341	52.6	4.8	15.71
			2	371	600	433	35.5		
		Ar	1	176	409	374	50.5	29.7	
	15	O_2_	1	162	374	342	53.5	4.5	16.04
			2	375	600	429	35		
		Ar	1	176	417	380	51.7	29.9	
	17	O_2_	1	154	376	349	56.7	4.1	15.92
			2	376	516	402	33.7		
		Ar	1	174	415	379	52.4	26.9	
Well 2	9	O_2_	1	157	376	332	49.6	9	13.21
			2	376	600	438	34.8		
		Ar	1	166	412	364	50.9	31.4	
	10	O_2_	1	156	376	330	56.8	7.7	14.24
			2	377	600	478	34.5		
		Ar	1	171	415	378	58.8	24.8	
	15	O_2_	1	152	364	323	53	9.2	13.93
			2	364	600	489	36.8		
		Ar	1	150	413	357	54.4	26.1	
	17	O_2_	1	146	380	295	53.4	7	13.84
			2	380	600	450	36.6		
		Ar	1	133	392	309	50.1	27.1	
	29	O_2_	1	150	371	335	49.1	9.8	15.23
			2	372	625	386	36		
		Ar	1	170	416	375	46.4	34.2	
	40	O2	1	172	364	329	37.3	17.9	13.97
			2	365	600	535	36.6		
		Ar	1	178	438	369	35.2	45.7	
	50	O_2_	1	160	378	338	55.4	5.7	15.38
			2	379	600	391	34.1		
		Ar	1	166	417	375	50.8	31.3	
	60	O_2_	1	160	376	336	53.3	8.5	15.05
			2	375	601	378	35.1		
		Ar	1	169	411	380	50.9	30.9	
	67	O_2_	1	164	382	339	53.5	7	15.65
			2	386	600	402	34.9		
		Ar	1	172	416	377	48.8	32.1	
	73	O_2_	1	151	381	340	67.4	15.2	18.86
			2	382	600	423	42		
		Ar	1	174	418	378	52.6	29.7	
Well 2	78	O_2_	1	151	376	337	55.2	5.6	15.39
			2	376	600	390	32.7		
		Ar	1	170	417	376	49.8	31.6	
Well 3	17	O_2_	1	174	380	338	27.9	48.2	7.71
			2	380	600	538	17.9		
		Ar	1	177	419	377	22.2	62.2	
	29	O_2_	1	149	378	337	55	9.2	15.61
			2	378	600	390	31.3		
		Ar	1	168	421	379	51.1	32.6	
	40	O_2_	1	161	382	339	54.3	9.7	14.94
			2	383	600	395	29.6		
		Ar	1	161	421	374	47.1	33.9	
	50	O_2_	1	161	381	336	42	31.6	11.17
			2	380	600	381	21.7		
		Ar	1	162	428	364	35.7	48.2	
	60	O_2_	1	151	380	340	55.8	6.7	15.402
			2	380	600	395	31.3		
		Ar	1	170	419	378	47.8	31.4	
	67	O_2_	1	150	372	337	44.1	22.1	10.966
			2	372	600	381	25.7		
		Ar	1	161	418	372	37.1	46	
	73	O_2_	1	162	377	335	49.5	15.5	14.457
			2	377	600	413	30.9		
		Ar	1	162	417	370	45.8	36.1	
	78	O_2_	1	155	378	338	56.4	8.5	13.761
			2	379	600	387	29.7		
		Ar	1	161	419	378	51.9	30.4	
	82	O_2_	1	158	372	335	50.9	10.6	14.483
			2	372	600	407	34.4		
		Ar	1	161	420	374	44.6	36.4	
	85	O_2_	1	158	380	336	62.1	4.7	13.102
			2	380	600	437	29.1		
		Ar	1	154	420	377	51.2	31.1	

Atm., atmosphere in which thermal analysis was carried out (oxygen or argon); number of main stages, the number of main biomass destruction stages that were determined by the number of main peaks on the differential scanning calorimetric curve (mW/mg). Imperceptible and weakly expressed stages were discarded in the analysis. t1 and t2, temperature of the beginning and end of the stage; tmax, temperature at which the maximum mass loss rate was observed for the sample; ∆m, total mass loss of the sample at this stage.

**Table 4 plants-11-01549-t004:** Comparison of thermal properties of BWW samples with analogues.

Sample	Source		Content, %					
		C	H	N	S	O	HHV, MJ/kg	Ash, %
BWW	Own research	46.2	4.99	0.44	0.21	48.1	14.85	11.7
*Furcellaria*	[76]	37.52	5.82	3.60	3.00	50.05	9.13	7.8
*Scenedesmus* sp.	[77]	46.3	6.81	3.28	0.28		21.5	7.0
Peat pellets	[78]	58.83	5.12	1.11		36.93	21.24	3.02
Coniferous wood	[78]	48.56	11.84	0.7	0.06	38.85	19.52	0.64
Poplar	[79,80]	51.60	6.00	0.60	0.02	41.70	18.3	3.77
Rice husk	[81,82]	49.40	6.20	0.30	0.40	43.70	15.72	3.98
Wheat straw	[78]	46.62	5.09	1.31	0.11	42.72	18.47	4.26

## Data Availability

Not applicable.

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
