# Peer review of "Feasibility of Old Bark and Wood Waste Recycling"

_plants, 2022, doi:10.3390/plants11121549_

Round 1

Reviewer 1 Report

Dear Authors,

Please find attached my evaluation of the manuscript.

Best regards

Author Response

Thank you so much for reviewing our paper. We really appreciate the tremendous work you have done in evaluating and correcting the text. We accept all your comments and suggestions and it really improved the quality of our paper.  Please see the attachment.

Best regards

Reviewer 2 Report

The manuscript entitled feasibility of old bark and wood waste recycling elucidated sustainable resources regarding wood processing, wood waste and recycling.The authors definitely addressed the properties of BWW of various storage periods are compared, the most common methods of processing and recycling of BWW are summarized, as well as the feasibility of using hydrothermal methods and composting for BWW processing. 

I think this is a useful article and I think it makes a potent scientific contribution to a sustainable resource. I would recommend the manuscript for major revision.

Just major points,

The introduction must be improved by incorporating more recent references including wood processing, wood waste and recycling.

Line 195, please address a concise table for analysis conditions.

In conclusion, please the contents detailed should be addressed including future scope and prospects in  feasibility of old bark and wood waste recycling.

Author Response

Thank you very much for your attention to our article. We really appreciate the tremendous work you have done. We accept all your comments and suggestions.  Please find point-by-point response in the attached file. 

Best regards

Round 2

Reviewer 2 Report

The authors have improved the manuscript for the publication.